# Integrating Environmental and Social Dimensions with Science-Based Knowledge for a Sustainable Pesticides Management—A Project of Lombardy Region in Italy

Maura Calliera [1,*], Andrea Di Guardo [2], Alba L'Astorina [3], Maurizio Polli [4], Antonio Finizio [2] and Ettore Capri [5]

1   Opera Research Centre, Università Cattolica del Sacro Cuore, 29122 Piacenza, Italy
2   Department of Earth and Environmental Sciences, University of Milano Bicocca, 20126 Milano, Italy; diguardo@iambientale.it (A.D.G.); antonio.finizio@unimib.it (A.F.)
3   Institute for Electromagnetic Sensing of Environment—National Research Council (IREA-CNR), 20137 Milano, Italy
4   Parco Regionale Adda Sud, 26900 Lodi, Italy; maurizio.polli@parcoaddasud.it
5   Department for Sustainable Food Process (DiSTAS), Università Cattolica del Sacro Cuore, 29122 Piacenza, Italy; ettore.capri@unicatt.it
*   Correspondence: maura.calliera@unicatt.it

**Abstract:** Achieving a change towards the sustainable use and management of pesticides requires a multiple perspective approach that combines traditional knowledge, experience of different local stakeholders, scientific expertise, and context-specific data to provide useful and understandable information for the target farmers. In this paper, the incorporation of the information on environmental and social dimensions into a "science-based" pesticide management practice is presented as an example of a replicable multidisciplinary approach. This approach depicts the importance of the context-specific scenario analysis and of the involvement of farmers starting from their practices and their knowledge. A diverse range of engagement initiatives have been adopted to consult, inform, and involve the community. Tools as target guidelines of good practices, self-evaluation checklists, and a user-friendly indicator that considers social, environmental, and territorial parameters of the specific area, gained a lot of interest and trust and have proven to be useful in disseminating the methodology of environmental risk assessment to farmers, supporting and assisting them in the comparison of different phytosanitary strategies at farm scale to identify weaknesses in their current pesticide management at farm level and to find corresponding corrective actions. The experience also highlighted the importance of the role of properly trained and informed advisors.

**Keywords:** sustainable pesticide use; management; engagement; check list; indicator

## 1. Introduction

The way we produce food and the agricultural practices that we follow can contribute to global greenhouse gas emissions, biodiversity loss, and environmental pollution; the way pesticides are managed might negatively affect the environment and generate conflicts. To achieve sustainable agri-food production, the recent trend of research and policy actions is to focus on the complex task of generating a long-term culture strategy that also includes a bottom-up adoption of sustainable and good practices [1–3]. In this framework, the evaluation of agricultural sustainability activities at the farm level has grown considerably [4–6], and several research programs, aimed to foster the adoption of sustainable practices with the involvement of farmers, have been developed in a multi-stakeholder's process [7–9]. The connection between farmers' socio-economic status and the perceived importance of sustainable agricultural practices is now well recognized and has also been considered in several European projects in the Horizon 2020 work program, such as BROWSE [10], WATERPROTECT [11], and INNOSETA [12].

Research and innovation also have a crucial role in future agricultural sustainability. The role lies in delivering security and competitiveness, providing technologies to support a sustainable intensification of the agricultural production, but also in improving the knowledge system where the traditional 'linear' model—from scientists to users—is gradually replaced by a more participative one. The participative model integrates production, advice, and education involving all actors: from farmers, to researchers, policy makers, companies, and citizens [13]. At the European level, several instruments and measures support the exchange of knowledge and promote the training and advice on innovation and technology. The farm advisory system (FAD) and the agricultural knowledge and information system (AKIS) highlight the importance of the farming community as a key player building a sustainable agri-system, including the sustainable use of pesticides, and stimulate a search for new approaches and design systems for an efficient knowledge transfer [14] (Indeed, sustainable pesticide management is a complex and dynamic process linked to a specific time and space, with the social dimension playing an important role due to interactions between several actors with different perspectives on sustainability. However, there are still very few existing empirical analyses of "farmers' perception" about pesticide good practices in real-world management contexts.

In this paper the incorporation of the environmental and social dimensions into a science-based pesticide management practice is presented as an example of a replicable multidisciplinary approach. This paper describes the experience gathered throughout a 4-year-long regional research project named TRAINAGRO, promoted in the framework of the European agricultural fund for rural development program (EAFRD), and aims to depict the importance of the context-specific scenario analysis and the involvement of farmers in the development of strategies towards the sustainable use of pesticides. A diverse range of engagement initiatives that have been adopted to consult, inform, and involve the community in the study area, and tools for enhancing the knowledge on the sustainable use of pesticide that consider social, environmental, and territorial parameters of the specific area, useful in disseminating the methodology of environmental risk assessment to farmers supporting and assisting them in the comparison of different phytosanitary strategies at farm scale will be described.

## 2. Materials and Methods

### 2.1. Area of Study

Italy represents an interesting case study for retroactive pesticide management and risk evaluation due to its agricultural heterogeneity and complexity.

Indeed, the Italian system is characterized by a highly decentralized national administration, where the government sets general rules and different regions (NUTS level 2) are expected to adapt and apply them in their territories according to the local environmental and socio-economic context.

In this framework, the Lombardy region is characterized by an agro-industrial system with a great market value that insists on a highly productive and extremely competitive territory and very high-income farms per work unit [15]. In this scenario, farmers must face the challenge to produce in a more sustainable way in the presence of several vulnerable areas. Although the monitoring data and the risk assessment of the water contamination by pesticides in the Lombardy region indicate, in general, a stationary situation [16], plant protection products still represent a critical point for surface and groundwater, particularly for the intensively cultivated crop types (maize and rice) that specifically require herbicide use [17]. In addition, given the high environmental value of Natura 2000 sites and protected natural areas, the regional authorities have deemed necessary to pay particular attention to the recognition of the risks deriving from the use of pesticides and, if necessary, to the identification of adequate mitigation measures.

For these reasons, the first two years of our activities were conducted in the local Parco Regionale dell'Adda Sud (PRAS) (focus area in Figure 1), 24,260 hectares, located along the lower course of the Adda river, a tributary of the Po river, with a length of about 60 km.

The PRAS hosts areas of outstanding natural beauty that preserve important biocoenoses linked to wetlands of considerable interest. One of the park's institutional purposes is to promote and restore the ecosystem services provided in a context where human activities, particularly agricultural, are intensive. Indeed, the park hosts different types of farms, as farms characterized by a highly mechanized and specialized agriculture, livestock farms, arboriculture activities, and agri-tourism. The main crop is maize. In this situation, the impact of agriculture and of pesticides on the environment and water could be of great importance.

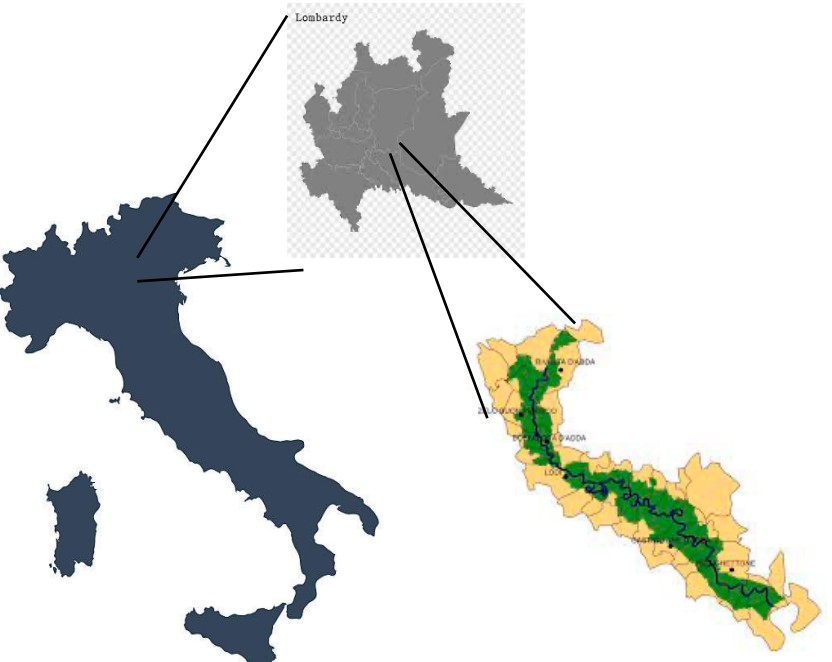

**Figure 1.** Study area. Parco Adda Sud and Lombardia Region.

Following the experience in the focus area, in the next two years, the activities of the project were extended to the plain part of the Lombardy region, in the area most densely cultivated with maize.

*2.2. Materials and Methods*

Following previous experiences aimed at understanding the 'life cycle' of the pesticides in the farm and at raising the community's awareness about preventive actions [18,19] we adopted a stepwise approach for the development of the project:

Step 1: Engagement activities in the study area and promotion of the interdisciplinary views;
Step 2: Development of tools for enhancing knowledge on the sustainable use of pesticides;
Step 3: Development of a context, farm-specific indicator for a comparative evaluation of different phytosanitary strategies and applicable mitigation measures at context level.

Below is the description of the methodology adopted for the implementation of the three respective steps.

2.2.1. STEP 1—Engagement Activities to Consult, Inform, and Involve, and Promotion of the Interdisciplinary Views

We started from the assumption that a complex socio-ecological issue, such as the sustainable use of pesticides, cannot be solved by just one actor but rather from a multi-actor approach. As stated in the introduction, sustainable agriculture is the result of complex "systemic interactions" between different subjects with diverse backgrounds and expertise. The challenge is to create conditions for interaction between them. Community engagement is increasingly encouraged as a method to improve project outcomes and, throughout the

4 years of the project duration, a diverse range of engagement initiatives/technics have been adopted to consult, inform, and involve the community [20], as summarized below and in Figure 2.

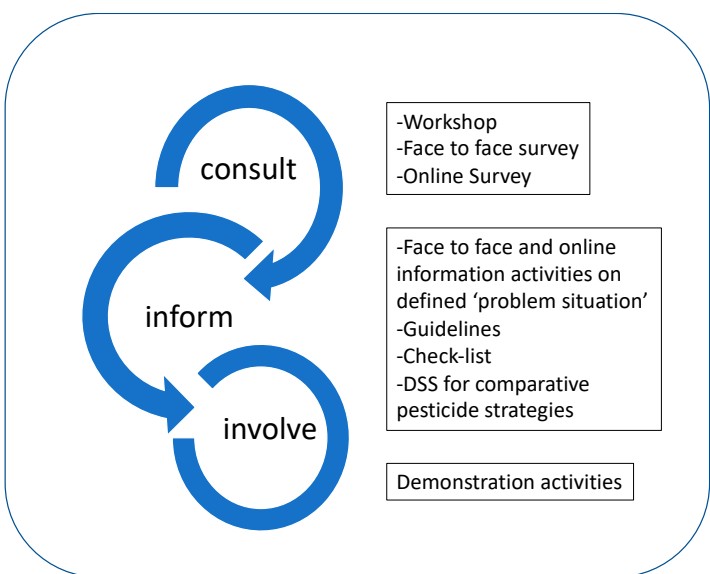

**Figure 2.** Flux of project's engagement activities.

Stakeholders Involved

In line with the objective of the measure of the rural development plan and to strengthen the links between research and practice, actors of different roles were involved to share skills and experiences. The participating speakers who contributed to fostering the dissemination of knowledge but also of innovative practices, products, and technologies for a sustainable use and management of pesticides were representatives of 6 universities, one of which in France, 8 crop protection companies, 3 companies involved in the production and distribution of technological innovations and agricultural machinery, experts representing the main trade union associations (industry and farmers), and professional associations. Public body representatives: experts from the Lombardy region phytosanitary service; Piacenza phytosanitary consortium; ALSIA—Lucana agency for development and innovation in agriculture; CAPA—professional center for training and services dedicated to agricultural workers; international centre pesticide safety (ICPS)-Sacco; Adda Sud Park; DG environment and climate of the Lombardy region; DG agriculture, food and green systems of the Lombardy region.

Consult—Initiatives That Seek Input from the Community for a Shared and Defined 'Problem Situation' and Build Policy Support

To tackle the health and environmental risks from an eco-systemic and long-term perspective, the diffusion of a culture of prevention and anticipation is considered the most appropriate tool This strategy assumes the knowledge of several context-specific elements: the various actors operating in the supply chain, the ways in which they behave in their daily practice, and the many factors affecting their decision to assume or not assume a sustainable behavior. While there is not a single 'best approach', different consultation strategies were organized, targeted to different groups of interest (farmers, advisors, representatives of the region, industry, . . . .) as described below.

1—*Workshop followed by a round table.*

This initiative was organized at the very first stage of the project. Opinions, current knowledge, available alternatives, and policy options for the improvement of the current pesticide management practices at the different farm scales of the local areas were discussed. Representatives of various stakeholder groups (pesticide industry associations, farmers associations, risk managers, and academia) were invited to the workshop, and expert

judgements were combined. This form of interaction by sharing data, judgements, and perception on the current state of knowledge helps in finding common ground. Indeed, the main goal of this approach was to resolve contradictory opinions and conflicts, to come up with a collective opinion, and to create trust in our project goals. An online (imposed by the COVID-19 pandemic restrictions) follow-up workshop with the same structure was then organized in the third year of the project.

2—*Exploratory Surveys.*

To calibrate the informative needs on the specific reference context, the project started with a preliminary phase of exploration about PPP management with the purpose of collecting information and bringing out the critical points and/or strengths for the sustainable use of PPPs at the farm level. In the study area, the on-farm life cycle of pesticides that were evaluated covered the following stages: transport, storage, mixture preparation, treatment, and remnant management.

Two exploratory qualitative surveys were launched to confirm the expert judgment and the trends and information shared and discussed at the round table. In total, 176 farmers were involved.

The first survey was conducted in 2018 in a face-to-face modality with the support of the 3 main local farmer associations. The analysis of the land use cartography [21] shows that 74% of the park territory is used for agricultural purposes, and maize represent 78% of the cultivated areas for a total of 14,115.79 ha. The average farm size area is 150 ha. The object of the survey was 100 randomly selected farms. In this questionnaire-based tool, farmers were requested to individually answer questions by choosing from a limited number of provided options. Although the investigation was passive, the tool represented an efficient way to obtain sufficient data in a short time. Following some expert reviews, 28 questions were included in the questionnaire, divided in two sections: (i) The on-farm life cycle of pesticides, and the critical points and/or strengths related to their sustainable management and use; (ii) The knowledge gap and information needs.

The second qualitative survey was conducted online, due to COVID-19 restrictions, in the year 2020; the aim was to confirm the trend of the first survey in the area that is most densely cultivated with maize in the region. This time, 3 more questions were added to assess farmers' level of interest in technological innovations in the agricultural field, their willingness to use new technologies, and the perceived barriers for the needed investment. The participants of the survey were 76 farmers.

Inform—Initiatives to Provide Input to the Community with the Aim to Inform, Educate, or Raise Awareness

Although there is already a substantial amount of knowledge available and agricultural research delivers new advancements, in line with the AKIS principles [22], a more perception-oriented and context-specific system is needed to achieve the European sustainable use directive goals [23]. Indeed, increasing research activities does not necessarily imply adoption of innovation, or change in behaviors, especially in agriculture where informative priorities differ between target groups, and the means of delivering the knowledge need to be sufficiently diverse, using flexible tools in order to be able to address all differences in knowledge and resources.

In our project, informative initiatives, described below, were planned, considering the results of consultation initiatives, which showed a growing interest in the adoption of more "modern" communication approaches, such as experimental, demonstrative, and participatory, and other appropriate techniques, which allow "learning through practice" and promote the understanding of the issues addressed.

1—*Informative days*. Several face-to-face and online "information days" were organized throughout the project period, focusing on topics highlighted by the farmers as the most interesting during the consultation phase, which are listed in the results section. The information days' agendas were planned for farmers, operators, consultants, and agronomists/technicians, and covered the different aspects of the sustainable manage-

ment of pesticides in the farm using a very practical approach to facilitate the exchange of experiences and skills between all participants.

2—*Tools to transform "contex data" into information.* A large part of the project was also dedicated to the design and planning of tools that facilitate the transformation of data into useful information for the farmers, such as checklists linked to guidelines for good agriculture practices and a decision support system, which through a newly developed impact indicator, allows the comparison of different phytosanitary strategies at the farm scale. Methods for the development of these tools are explained in Sections 2.2.2 and 2.2.3.

Involve—Initiatives That Build Active and Connected Communities

The project organized demonstration activities (described in the results section, Section 3.1.3) with the aim of educating and building trust and effective long-term relationships between stakeholders involved in the pesticide management. With this aim, best management practices to sustainably manage pesticides at farm level were demonstrated; the final objective was to limit or prevent point source contamination and give context-specific and applicable information about pesticide risk mitigation measures to prevent diffuse contamination, as requested by farmers in the consultation phase. The target group of this initiative was mainly farmers but also advisors. Indeed, as confirmed by the last European Commission evaluation of the common agricultural policy (CAP)'s impact on knowledge exchange and advisory activities "[ . . . ] *there is also a need to update advisers' knowledge and skills. Agricultural advice is an essential lever to change farming practices and providing qualified and impartial advisory services remains an important issue*" [24].

2.2.2. STEP 2—Development of Tools for Enhancing the Knowledge on the Sustainable Use of Pesticides

Based on the data collected in the surveys and their analysis and following previous experiences, the following tools were developed:

(i) Operational guidelines of good practices: an operational guideline for the sustainable use of pesticides available for training and raising awareness among professionals, with recommendations for the responsible, safe, and sustainable use of these products;

(ii) Free online software to identify the critical issues on the farm, taking into account both the structural aspects of the farm and the behavior of the farmer (checklists) to assist the user with the analysis of their own agricultural practices and to help them identify critical points in pesticide management.

The structure of the online checklist follows the chapters of the guidelines; the user has to answer several questions that cover the main topics of each chapter. A weighting system is used for each choice in order to build a set of indicators. Please refer to Calliera et al., 2013 [18] for a better description of the checklist design.

2.2.3. STEP 3—Development of a Context- and Farm-Specific Indicator for a Comparative Evaluation of Different Phytosanitary Strategies

A novel GIS user-friendly tool (MIMERA: mitigation measures and environmental risk assessment) [25] has been developed to disseminate the use of the environmental risk assessment (ERA) methodology to farmers, so that they could easily understand the impact of pesticide strategies adopted and build different scenarios for testing alternative and more sustainable pest management strategies. The tool incorporates the entire set of environmental and territorial parameters of a specific area (cadastral parcels, pedology, meteorological data, slope, crop data, and physical-chemical and ecotoxicological properties of commercial pesticide formulations registered on selected crops). In this way, the user focuses on well-known information (crops and pesticide strategies adopted in each field) and he/she can select the best pesticide management strategy according to the environmental characteristics of farm parcels. In addition, MIMERA suggests the efficiency and applicability of risk mitigation measures to protect surface water bodies near the treated areas. It also allows tracking of temporal pesticide risk trends for self-evaluation and

communication of progress in reaching the reduction goals of pesticides' environmental impact in farms.

## 3. Results

### 3.1. Step1

#### 3.1.1. Engagement Activities to Consult

*1—Round table combining individual expert judgements.* Selected participants with expertise in different disciplines were invited. The selected experts were representative of the study area group of policymakers, industry, farmers' associations and academia dealing with, or subject to the consequences and the implementation of the sustainable use directive. The topic addressed was the correct use of plant protection products to reduce the risks and impacts associated with the use of these products and encourage the adoption of innovative technologies and the adoption of sustainable behaviors. Experts were allowed to an open discussion managed by a moderator encouraging participants to express their views and debate on the issues. The meeting was held under the Chatham House Rule [26], by which participants are not quoted in the minutes and so are given more freedom to speak.

Thoughts arising from the round table, and summarized below, were taken as a basis for the survey

- It is necessary to strengthen the link between agriculture and research;
- The farmers of the reference area are trained, but the training is disconnected from reality;
- In the study area, the transmission of information is "relational" and certain figures, such as the contractor and the consultant/extensor, are important;
- Information on the sustainable use of pesticides must be context-specific;
- Avoid the traditional top-down approach.

*2—Explorative Qualitative Surveys.* This survey was launched to confirm the expert judgment, the trends and information shared and discussed at the round table regarding the study area. The frequency of the observations was analyzed using Microsoft Excel. The results of the two different surveys are substantially consistent, as can be seen in Table 1 for the main questions.

**Table 1.** Exploratory surveys.

|  | Survey 1 | Survey 2 |
|---|---|---|
| Number of Respondent | 98 | 76 |
| Male | 84% | 87% |
| Phytosanitary treatments carried out | | |
| by the farm manager | 44% | 54% |
| by contractor | 29% | 17% |
| Decision taken by | | |
| own experience | 26% | 27% |
| private technician/advisor | 27% | 41% |
| retailer | 16% | 12% |
| Age of sprayer (excluding blank responses and contractors) | | |
| 0–3 years | 18% | 14% |
| 3–5 years | 32% | 24% |
| 5–10 years | 30% | 37% |
| More than 10 years | 13% | 25% |
| In-farm sprayer cleaning: | | |
| External sprayer cleaning | 60% | 76% |
| Internal sprayer cleaning | 58% | 46% |

*Social dynamics and behaviors.* What specifically arises is that the rural reality we have explored is mainly managed by male farmers (84% in the first survey and 87% in the second survey). This trend is also confirmed by the national statistical data [27].

The analysis of the responses confirms the expert round table conclusion and clearly highlights two important figures who are responsible for carrying out the treatments: the owner (responsible for the treatments in most of the cases) and the contractor (playing an important role in our study area, despite the farm size, as confirmed by the second survey). Demonstration and information activities must, therefore, include or involve this important category as well. Regarding the farmers deciding to carry out a phytosanitary treatment, apart from their own experience (26%), the most important professionals for delivering information and support to farmers are private technicians/advisors (27%) and retailers (16%). Advisors are considered to be trustworthy, and therefore, their involvement (and training) in the project becomes very important in order to raise awareness. In the first survey, conducted in the Parco regionale dell'Adda Sud, the option "consulting neighboring" reaches 12% of the responses, confirming the community character of the agricultural reality of reference. Only few farmers have declared referring to the bulletins. This trend is confirmed by the second survey that once more highlights the importance of the role of technician/advisor (preferred by 41% of respondents)

Respondents can be considered adequately informed, confirming the experts' judgment. Indeed, 73% of them are authorized to purchase and use pesticide. Since the approval of the National Action Plan (as stated in the EC directive 128/2009), rules linked to the training of professional users were changed, and from year 2015, the qualification certificate became a mandatory requirement. The second survey, with participants located in the Lombardy region, confirms the high educational qualification of the participants (40% high school graduates and 20% university graduates).

*Equipment and calibration.* The age of sprayers used for phytosanitary treatments is another critical point. The sprayers used by the majority of the farmers are relatively old; less than 50% of all sprayers in the surveyed sample are less than 5 years old. On the other hand, the "Cost of Crop Protection measure" study of the European Parliamentary Research Service [28] highlights that replacing equipment with the latest technology is not economically and environmentally viable when the existing equipment is less than 10 years old. For this reason, the calibration of the existing sprayers has been the subject of demonstration activities. Demonstrating the proper use of existing machineries could help to ensure the correct sprayer operation in order to limit the drift as well as point to source contamination events caused by dripping and loss.

*Point source pollution.* Point source pollution and water waste management are the two main challenges for the sustainable use of pesticides. The cleaning of the sprayer is one of the most important aspects, and it is generally performed in the farm (58% of positive responses for Survey 1 and 45% for Survey 2), in particular in large-sized farms. Resulting waters, if not properly managed, can be a source of environmental contamination. The external washing of the sprayer is mostly carried out in farms (60% of positive responses for Survey 1 and 70% for Survey 2). If not correctly managed, these waters, although very diluted, can be a source of environmental contamination, in particular if operations are always carried out in the same place [29].

*Interest in technological innovations.* The degree of interest in technological innovations in the agricultural field is very high for the majority of respondents (Figure 3), but it is influenced by the investment needs. In detail, 30% of the respondents are interested but prefer to wait for their adoption until costs are at the level of already available technologies, 13% prefer to wait until they are widely spread, 27% are very interested and are constantly informed with specialized magazines (on the web and/or on paper), and 9% are very interested and in connection with research institutions to carry out technological trials in the farm. This last figure, small in percentage terms, is particularly interesting when compared to the size of the company; larger farms are the most interested in applying innovation and are ready to invest more than the others.

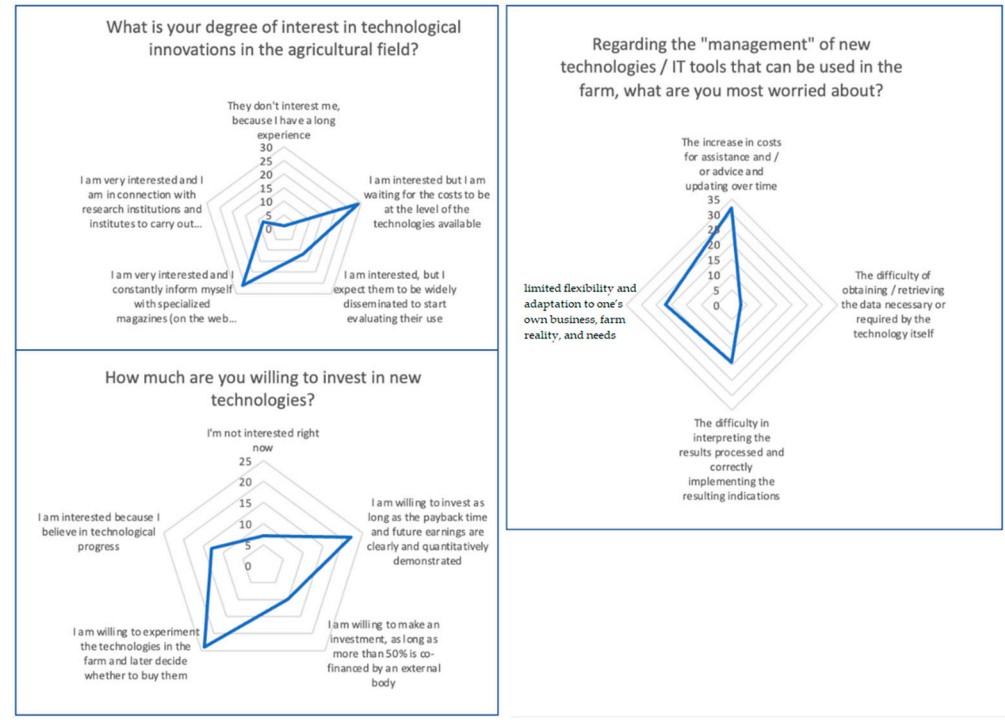

**Figure 3.** Interest in technological innovations.

Answers to the question "How much are you willing to invest in new technologies" were clearly linked to the technologies' usefulness both in terms of efficiency and economic viability. Indeed, 32% of respondents are willing to test the new technologies and then decide whether to buy or not, 29% of respondents are willing to make an investment as long as the payback time and future earnings are clearly and quantitatively demonstrated, while 13% of them are willing to make an investment as long as at least 50% of the total cost is co-financed by an external body.

With respect to the "management" of new technologies or IT tools that can be used in the farm, the aspect that most worries the respondents (42%) is the increase in costs for the assistance and consultancy and updating over time, followed by limited flexibility and adaptation to one's own business, farm reality, and needs (29%). Among the main difficulties there is the interpretation of data/results.

This information is provided only by the respondents of the second survey and is of qualitative nature. However, they confirm a trend highlighted by a survey conducted at a national level [30,31].

### 3.1.2. Engagement Activities to Inform

*Informative days.* In total, eight events were organized. In 2018, four information events were organized from March to November at the Parco Regionale dell'Adda Sud, which addressed the main aspects related to the sustainable management of pesticides, such as: (i) The need to strengthen the link between agriculture and research; (ii) The digitalization and managing farm data; (iii) Reading the PPP label information and its meanings; (iv) The protection of biodiversity, the ecosystem services, and the management of sustainability within the park. A total of 108 farmers and technicians participated in the events.

In 2021, four online information events (due to COVID-19) were organized, and a total of 579 participants signed up for the events. This group of events was titled "Transforming farm data into information". The following topics were addressed: (i) Information needs from different points of view; (ii) Agriculture 4.0 and digital tools to support phytosanitary strategies at the farm level; (iii) Addressing the pest resistance to pesticides; and (iv) The meaning of the information provided in the PPP label. The events were videorecorded, and 23 informative videos about the issues addressed were obtained.

Apart from the first event, which was the project kickoff event, the technical topic that gained the highest interest was the digitalization and Agriculture 4.0 topic, addressed from a practical point of view. Variation in the number of participants over time was observed and was mainly linked to the following factors:

A—Temporal. At the beginning of each year, agricultural operators were more willing to participate in information and updating activities, as at that period they were less involved in field operations;

B—Spatial. The attribution of training credits helped to gather a high number of participants residing in other regions especially in the online events. In the events following the first, the participants were mostly from the Lombardy region, the target area of the project;

C—COVID-19 has affected the learning experience. The transition to the online format worked remarkably well, with a high level of participation, and, in turn, addressed some concerns raised by the temporal factor. The speaker presentations were published and made available at the end of each event on the website www.trainagro.it, which can be found in the documents section (accessed on 27 March 2023).

### 3.1.3. Engagement Activities to Involve

*Demonstration activities*. During the first step of the project, nine demonstration activities were organized in farms located within the boundaries of the Parco Regionale dell'Adda Sud, with a total of 210 participants. The topics covered were: (i) Point source pollution prevention and wastewater management; (ii) Regulation and calibration of sprayers as criteria for choosing sprayers and distribution volumes; (iii) Effectiveness and efficiency of drift mitigation measures and analysis of different types of anti-drift nozzles; (iv) Effectiveness and efficiency of surface runoff mitigation measures; (v) Use of drones for corn treatments with low environmental impact products.

Six more demonstration activities were organized in the year 2022 on the same topics except for run-off, a phenomenon considered of lesser importance for pesticide water contamination, given the morphology of the territory under analysis. The first two of these demonstrations were available also via online streaming. Demonstration activities were also conducted during the COVID-19 pandemic, but in this case, the lack of interaction was identified as the biggest disadvantage compared to the "in presence" or face-to-face events. Two demonstration activities were planned in autumn, a period requested by the farmers themselves. The number of participants was in total 190.

Despite the demonstration activities being targeted at specific farm conditions, representative of the surrounding realities, the farmers had to travel to view the demonstrations, and did not apply the methods themselves, but advisors present assisted the farmer with applying the method to their own farms.

This reaffirms the importance of the role of the advisors, who must be properly trained and informed. On the other hand, advisors' knowledge transfer should mainly be based on user orientation, taking into account individual needs and the adequacy of the content and involve farmers in making decisions that affect them. Furthermore, the planning of activities and the type of innovation demonstrated need to be affordable by the target farmers.

### 3.2. STEP2 and STEP3—Development of Tools for Enhancing Knowledge

Guidelines were developed by the working group, following the results of the surveys and also considering documentation already produced, in particular, the documents developed under the Project Life 'Training the Operators to Prevent Pollution from Point Sources' TOPPS.

The guidelines cover the on-farm life cycle of pesticides from the purchase and transportation to the farm, application, and disposal of waste water to the management of empty containers; each step is developed in a specific chapter. Simple and intuitive language was chosen with explanatory drawings and tables, and text and practical advice were provided taking into account:

- Structural issues—such as pesticide storage, sprayers, structures for preparing mixtures; and
- Behavioral issues—best practices to encourage the correct behavior of the operator/worker.

A hypertext version of the guidelines and the online checklist to self-evaluate the current farmer level of sustainable use of pesticide guidelines were provided in the free online website (www.trainagro.it, accessed on 27 March 2023) to assist farmers. For more effective communication, we have chosen a gauge for the graphical representation of the results of the checklist indicators (Figure 4).

In the free online project website, the indicator MIMERA is available. Users can select the best pesticide management strategy according to the environmental characteristics of the selected farm parcels, as shown in Figure 5.

Since the first implementation, guidelines of good practices and self-evaluation checklists received a large interest from stakeholders, in particular technicians from extension services and farmers. During the period from January 2020 to August 2022, the public website, which hosts the Guidelines web pages, acquired 9662 total users, 87% of which were new users and 13% returning visitors. The website pages have been visited 24,436 times, with an average of 2 pages per session. During the same period, the private website (which stores self-evaluation checklists and the MIMERA indicator) acquired 318 users, the 76% of which were new users and 24% returning visitors. The average session duration per user was 10 min and 50 s, and the average number of pages visited per session was 10.43. These data testify that the public website gained a large interest as an informative medium about the project activities, while only a fraction of the visitors visited the Guidelines web pages. On the other hand, a much smaller number of users visited the private site, but they spent more time on checklists and MIMERA, testifying their interest in the two tools.

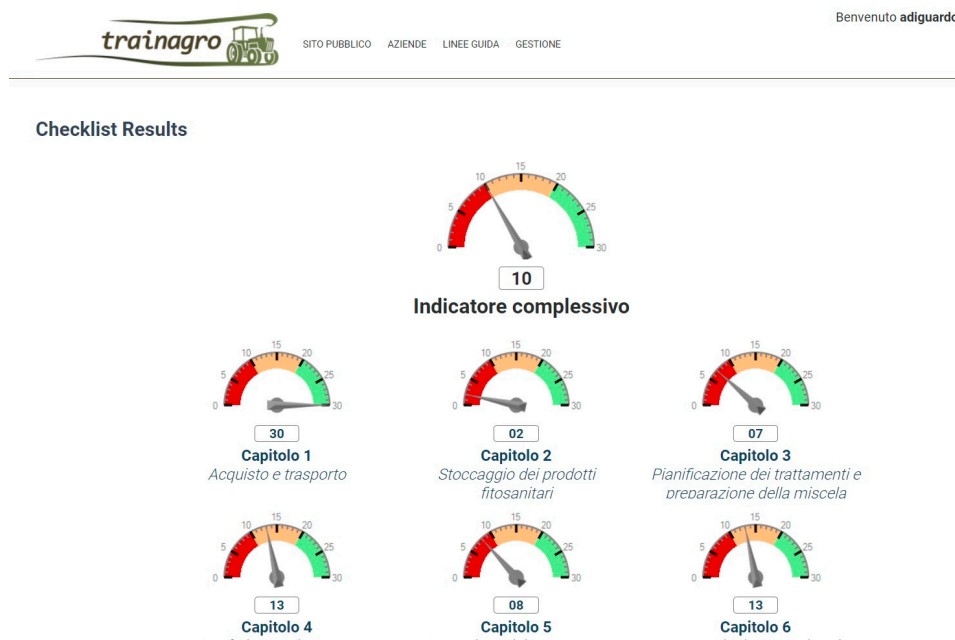

**Figure 4.** Graphical representation (gauge) of the check list indicators results.

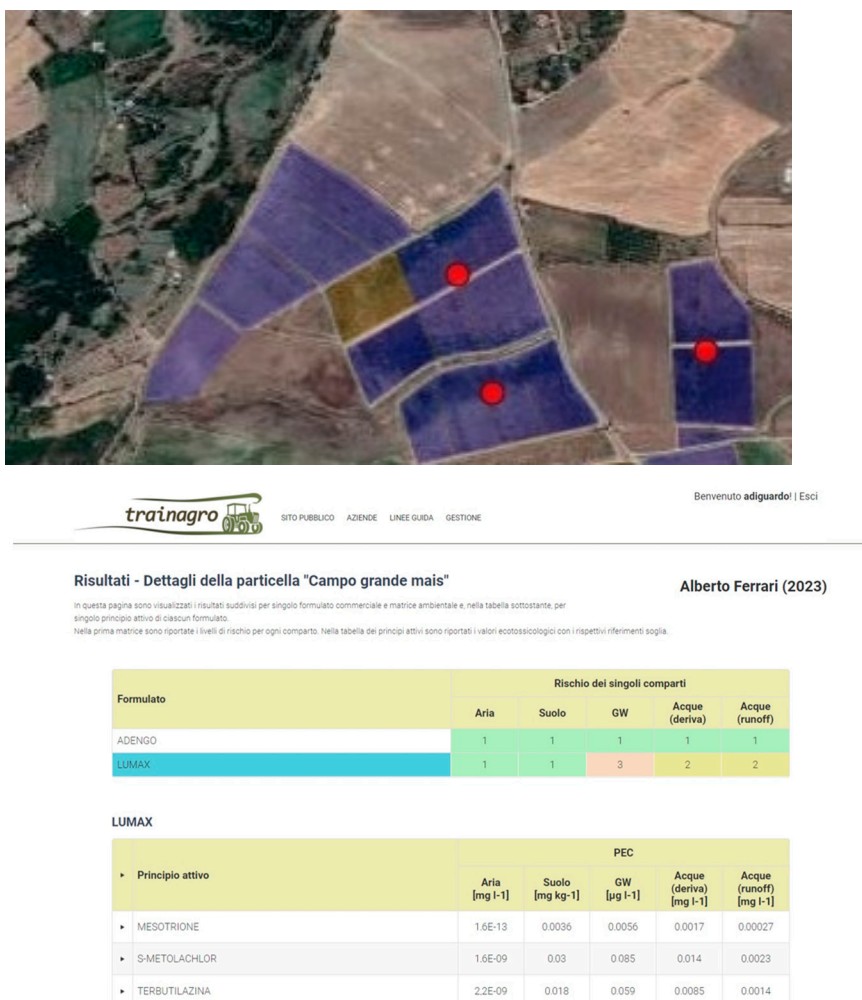

**Figure 5.** Example of MIMERA indicator application and results.

## 4. Discussion

Despite the advancement in research and the innovations available, an effective transition towards the sustainable use of pesticides could not be expected without the consideration of the farmers' needs and their social context. Additionally, if farmers in the study area are trained, information in agriculture aiming towards a sustainable use of pesticide must be addressed, taking into account that it is important to understand the social dynamics of the context, factors affecting farmers' decision to assume or not assume a sustainable behavior and which are the professional figures that the farmers rely on to obtain the information. To better transfer knowledge and experiences, the reputation and commitment of those who are recognized as "competent" are critical. Information providers must be able to provide information and solutions through demonstration and other methods that are adequate for the existing needs and at the same time effective.

As an example after the demonstration activities on the use of drones, the corn area treatments with low environmental impact products against Ostrinia nubilalis increased in the Adda Park area, as confirmed by the contractor.

Extension often plays a key role in farmers' pest management choices and new technology.

Round tables and explorative surveys at the beginning of all the activities with the involvement of the various actors operating in the supply chain have proven to be effective tools for obtaining information and building relationships. The pandemic caused by COVID-19 has changed the way information is provided, and the online information events have had a good response in terms of numbers of participation, demonstrating that this tool no longer provokes mistrust in our study area. Demonstration activities, on

the other hand, are less effective without in-person meetings. In general terms, both for the informative events and the demonstrative ones, a high-level participation is linked to the correct evaluation of the timings and places of organizations. Target guidelines of good practices, self-evaluation checklists, and the GIS user-friendly indicator gained a large interest and trust while incorporating social, environmental, and territorial parameters of the specific area and helping to disseminate the environmental risk assessment (ERA) methodology to farmers supporting them in identifying the flaws in their current pesticide management at farm level, as well as their corresponding corrective actions.

## 5. Conclusions

Based on the obtained results, the following statements can be made:

The adoption of measures for a sustainable use of pesticide varies between regions and crops and, as confirmed also by the cost of crop protection measure study of the European Parliamentary Research Service [28], the adoption of new techniques depends on farm size, the availability of budget for investments, and the risk attitude of farmers, since the application of new technologies creates uncertainty about efficacy. Therefore, achieving a change requires combining traditional knowledge, experience of different local stakeholders, scientific expertise, and context-specific data able to provide affordable useful and understandable information to the target farmers.

Our experience reaffirms the importance of the role of the advisors, who must be properly trained and informed [32]. The exchange through the organization of interactive moments is an opportunity to directly learn from other advisor/extensor and researchers and is considered very effective in the introduction of innovation at the farm level.

Knowledge dissemination activities should be provided balancing bottom-up and top-down approaches, considering multiple perspectives, working towards agreements between stakeholder groups, such as farmers' associations, advisors, researchers, policymakers, industry, and the local communities. Recognition of all farmers, including smallholder farmers, in a long-term perspective must be ensured. The important lesson we learned is that parties involved in pesticide use sustainability (but also in sustainability in general), should be prepared to promote a dialogue instead of a debate, as also recently recommended by the European Commission [33]. This means considering many possible right answers, perspectives, and behaviors that are functions of the analyzed context-specific scenario. At the European level, the Member State Strategic Plan developed under the CAP will have to take this into account to achieve the goals of the European Green Deal and the Farm to Fork Strategy.

**Author Contributions:** Conceptualization, M.C. and A.D.G.; methodology, M.C. and A.D.G.; writing—original draft preparation, M.C.; writing—review and editing, M.C., A.D.G., A.F., E.C., A.L. and M.P.; project administration, E.C. and A.F. All authors have read and agreed to the published version of the manuscript.

**Funding:** The work has been funded by the Lombardia Region (Italy) "Progetto TRAINAGRO2020—ammesso al finanziamento con d.d.s 7177 del 19 giugno 2020 PSR 2014–2020 della Lombardia. Operazione 1.2.01—Progetti dimostrativi e azioni di informazione—d.d.s. 11791/2019".

**Institutional Review Board Statement:** Not applicable.

**Informed Consent Statement:** Informed consent was obtained from all subjects involved in the study.

**Data Availability Statement:** For data supporting reported results please contact the corresponding author.

**Acknowledgments:** This paper was produced also in the frame of Ph.d. in Agro-food system (AGRISYSTEM)-XXXVIII Cycle. We also want to thank Anastasia Lomadze for the English revision of paper.

**Conflicts of Interest:** The authors declare no conflict of interest. The funders had no role in the design of the study; in the collection, analyses, or interpretation of data; in the writing of the manuscript; or in the decision to publish the results.

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
