# Peer review of "Integrating Environmental and Social Dimensions with Science-Based Knowledge for a Sustainable Pesticides Management—A Project of Lombardy Region in Italy"

_sustainability, doi:10.3390/su15107843_

Round 1
Reviewer 1 Report
I found this paper really interesting and interdisciplinary in terms on how to address the issue of pesticide use in a large and important agricultural area in northern Italy, involving all the stakeholders from farmers to industry. I missed the issues of consumer needs and the alternatives of not to use pesticides at all, under the current agricultural practices, but I understand the complexities on dealing with this issues. Probably it is not feasible under the current circumstances, but setting the scenario of non -using pesticides may also be considered an alternative . I think the consideration and analysis of farmers needs is a keystone in the approach used and to provide them with simple tools such as the MIMERA tool looks very straightforward. I think that one of the merits of this paper is to test both a bottom up as well as a top down approach simultaneously, considering the pandemic restrictions to do field research like this. I find the methodology scalable and that may be applied on a wide variety of farming situations across the globe since consider a dialogue for sustainability that to my understanding is one of the possibilities to overcome the current environmental crisis. Finally, it would be great if the authors may mention some of the outcomes of the research in terms of pesticide management in this specific context, for instance in terms of decisions to use less toxic chemicals.My recommendation is of course to accept this manuscript.
Author Response
Reviewer 1
We thank the reviewer for the appreciation of our paper.
1-Regarding the outcomes of the research in terms of pesticide management in this specific context, we add in the "Discussion section" that “as an example of impact after the demonstration activities on the use of drones, the corn area treatments with low environmental impact products against Ostrinia nubilalis, increased in the Adda Park area as confirmed by the contractor.
2-The objective of the overall work was to improve the sustainability of pesticide use and to limit and prevent environmental contamination, so, although we are aware of the need for a shift towards an almost partial divestment of pesticides, the work refers to those directives that aim to reduce the effects of point and diffuse contamination and for this we confirm that setting the scenario of non -using pesticides was not feasible
Reviewer 2 Report
The title corresponds to the content of the paper.
The aim of research is clearly and fully pointed.
The key words are appropriate.
- Scientific methodology is applied correctly for this type of study.
- Results are clearly presented and discussed.
- Tables, figures, pictures are clear.
The socio-demographic section can be presented in a table in a more concise manner
Including sub-chapters would guide the reader through the manuscript better. In addition comparison with the literature is largely missing.
A literature review is incomplete. It would be expected that the state of the art literature on sustainable pesticides management references must be improve, I recommend to borrow from other countries that are similar in terms of their sustainable pesticides management.
Author Response
We thank the reviewer for the opportunity given to improve our work and for the suggestions.
As requested,
1- in the chapter 3 result, to better explain the socio-demographic section we provide in the text the table 1.
2-We changed the chapter 2 and divided the text in sub-chapter as suggested. Below the changes
- Materials and Methods
2.1 Area of study
2.2 Materials and Methods
2.2.1 STEP 1 - Engagement activities to consult, inform, and involve and promotion of the interdisciplinary views
2.2.1.1- STEP 1 - Stakeholders involved.
2.2.1.2 -STEP 1 - Consult. Initiatives that seek input from the community, for a shared and defined ‘problem situation’ and build policy support.
2.2.1.3 STEP 1- Inform. Initiatives to provide input to the community with the aim to inform, educate, or raise awareness.
2.2.1.4 STEP 1-Involve Initiatives that build active and connected communities.
2.2.2 STEP 2 - Development of tools for enhancing the knowledge on the sustainable use of pesticides.
2.2.3 STEP 3 - Development of a context, farm specific indicator for a comparative evaluation of different phytosanitary strategies
3-We improve our literature references as requested.